# Improving Reproducibility to Enhance Scientific Rigor through Consideration of Mouse Diet

**DOI:** 10.3390/ani12243448

**Published:** 2022-12-07

**Authors:** Cara J. Westmark, James Brower, Patrice K. Held

**Affiliations:** 1Department of Neurology, University of Wisconsin, Madison, WI 53706, USA; 2Molecular Environmental Toxicology Center, University of Wisconsin, Madison, WI 53706, USA; 3Wisconsin State Laboratory of Hygiene, University of Wisconsin, Madison, WI 53706, USA

**Keywords:** scientific rigor, mouse diet, C57BL/6J, Teklad 2019, AIN-76A, Purina 5015

## Abstract

**Simple Summary:**

Our current inability as a biomedical field to reproduce and replicate published scientific findings is a major crisis. A contributing factor is lack of reporting of experimental variables in scientific publications. For example, rodent diet is rarely reported unless the research regards a dietary intervention. Our prior research found significant differences in seizure susceptibility and growth anthropometrics in mice as an effect of diet. Herein, we compare growth, behavior and blood biomarkers in male and female mice as an effect of three common rodent diets. We found significant differences in body weight and blood amino acid levels in the most commonly studied strain of mice. These findings contribute to a growing body of knowledge regarding the role of diet in health and disease as well as the need for detailed reporting of experimental variables, including diet, in scientific publications and presentations.

**Abstract:**

Animal husbandry conditions, including rodent diet, constitute an example highlighting the importance of reporting experimental variables to enhance scientific rigor. In the present study, we examine the effects of three common rodent diets including two chows (Purina 5015 and Teklad 2019) and one purified ingredient diet (AIN-76A) on growth anthropometrics (body weight), behavior (nest building, actigraphy, passive avoidance) and blood biomarkers (ketones, glucose, amino acid profiles) in male and female C57BL/6J mice. We find increased body weight in response to the chows compared to purified ingredient diet albeit selectively in male mice. We did not find significantly altered behavior in female or male wild type C57BL/6J mice. However, amino acid profiles changed as an effect of sex and diet. These data contribute to a growing body of knowledge indicating that rodent diet impacts experimental outcomes and needs to be considered in study design and reporting.

## 1. Introduction

Transparent experimental variables are essential to accelerate scientific progress, which hinges on our ability to reproduce prior findings. Landis et al. published an article in Nature in 2012, “A call for transparent reporting to optimize the predictive value of preclinical research,” summarizing recommendations by major stakeholders in response to a National Institute of Neurological Disorders and Stroke (NINDS) workshop discussing how to improve methodological reporting of animal studies in grant applications and publications [1]. The most important element required for scientific reproducibility is reporting studies in sufficient detail to allow others to evaluate the previous findings and repeat them. 

Environmental and genetic variables associated with animal husbandry likely contribute to data irreproducibility between laboratories. Others have found significant differences in mouse longevity in response to vivarium noise levels [2] as well as behavior in response to mouse strain background [3]. Diet is a good example to illustrate the importance of reporting experimental variables to enhance scientific rigor. Bioactive components in diets could interact with test drugs, which may help explain why many pharmaceutical agents are very successful in preclinical studies where drugs are tested in animals maintained on single-source diets, but do not function well in the clinic where patients eat a varied diet.

A serendipitous finding in our mouse colony changed how we think about rodent diets. We were conducting drug efficacy studies to attenuate seizures in mouse models of neurological disease. Daily restraint of the mice for abdominal drug injections is stressful to the animals, so we incorporated the drugs into the mouse feed. Since multiple batches of feed would be needed and we wanted to control as many factors as possible, we investigated the constituents of the standard vivarium feed, which at the time was Purina 5015. Inexpensive chows, such as Purina 5015, contain cereal grains including wheat and soy, animal by-products, and variable levels of non-nutrients such as phytoestrogens, toxic heavy elements, and pesticides. The proprietary formulas are secret and vary over time. Thus, we incorporated our test drug into an open formula, purified ingredient diet for rodents. Purified ingredient diets are highly refined with minimal batch-to-batch variability. To our surprise, the control purified ingredient diet, with no drug added, reduced seizure propensity in the mice by 50% compared to Purina 5015. This serendipitous finding ignited our interest in nutrition, and a significant focus of our research over the past decade has aimed at understanding the role of diet on neurological outcomes in mouse models of autism as well as in humans with autism spectrum disorders (ASD). 

The researchers in our laboratory have found that standard grain/soy-based vivarium chows, in comparison to purified ingredient diets, alter behavior and increase weight gain in mice, and more so in developmental disability models [4,5,6]. These original findings have been followed up in humans through retrospective analysis of medical records with findings that children with autism who were fed soy-based infant formula were more likely to exhibit seizures and other comorbidities [7,8,9]. These data served as the premise to conduct a case–control survey study evaluating the impact of early childhood feeding on the severity of common fragile X syndrome (FXS) phenotypes using the Fragile X Online Registry With Accessible Research Database (FORWARD), a national registry of FXS families. The project included collaboration with 10 FXS clinics across the U.S. and the data indicate that children with FXS who were fed soy-based infant formula had higher comorbidity of autism, allergies, and gastrointestinal problems [10,11]. The findings have significant implications for babies who are fed soy-based infant formula and could justify newborn screening for FXS if an early dietary intervention reduced disease comorbidities. It remains to be determined if soy is a cause or a consequence of more severe comorbidities in FXS and autism. Ongoing work in the laboratory is testing the effects of exactly matched casein- and soy-based diet on seizures, behavior, and biomarkers in *Fmr1^KO^* mice. 

Given the high use of single-source diets in biomedical research, it is important to examine the metabolic and physiological effects of commonly employed diets. Here, we aim to compare the effect of three commonly used mouse diets on growth, behavior and blood biomarkers in male and female, wild type C57BL/6J mice, which is the most widely used inbred mouse strain for preclinical research. We find significant differences in body weight and blood amino acid levels dependent on diet.

## 2. Materials and Methods

Mouse Husbandry: C57BL/6J mice were obtained from Jackson Laboratories (catalog #000664) at 3 weeks of age; 24 females and 21 males. The C57BL/6J background strain has been extensively utilized for nutrition and neuroscience research. Mice were housed in microisolator cages on a 12 h (0600–1800) light cycle with ad libitum access to feed and water (specific diets specified below). The bedding (Shepherd’s Cob + Plus, ¼ inch cob) contained nesting material as the only source of environmental enrichment. The animal study protocol was approved by the Institutional Review Board of the University of Wisconsin, Madison (protocol code M005224).

Diet Formulations: Purina 5015 mouse diet is a complete life-cycle diet for the reproduction, growth, and maintenance of mice (LabDiet, St. Louis, MO, USA) with a metabolizable energy density of 3.70 kcal/g and main ingredients of ground wheat, dehulled soybean meal and ground corn (Table 1). Teklad 2019 is a fixed formula, extruded diet with an metabolizable energy density of 3.3 kcal/g and main ingredients of ground wheat, ground corn, corn gluten meal and wheat middlings (Envigo, Fitchburg, WI, USA). Teklad 2019 does not contain any alfalfa, soybean meal, animal protein or fish meal, which minimizes the occurrence of natural phytoestrogens and nitrosamines. AIN-76A is a purified ingredient diet with an energy density of 3.79 kcal/g and ingredients of sucrose, casein, corn starch, corn oil, cellulose, mineral mix, vitamin mix, DL-methionine and choline bitartrate (Bioserv, Flemington, NJ, USA).

Study Design: Mice were received from Jackson Laboratories, weighed, and tested in a neuroassessment battery at postnatal day 21. Females (*n* = 8 each) and males (*n* = 7 each) were randomized to three test diets (Purina 5015, Teklad 2019 and AIN-76A). The number of mice tested was based on prior behavioral experiments. Mice were weighed weekly for four weeks and then biweekly, and tested in nest building, actigraphy, and passive avoidance assays (Figure 1, study design). 

Mouse Behavior: Mice underwent an abbreviated Irwin murine neurobehavioral screen at day 1 and day 56 of diet treatment as previously described [12]. Assessments included: body weight, tremors, eyelid closure, piloerection, lacrimation, salivation and dirty coat at both time points plus grip strength and righting reflex at the second time point.

Nest building was conducted as previously described [13]. Nest building is a paradigm for activities of daily living and well-being in mice [14,15]. Briefly, corncob bedding and two 2 × 2″ white cotton nestlets (catalog #NES3600; Ancare, Baltimore, NY, USA) were added to clean standard cages with feed and water. A single mouse was transferred to a test cage at 3 pm. At 7 am the following day, the mouse was returned to its home cage. The empty nestlet cage was photographed and the length, width, and height of the nests were measured. Nests were scored based on the scale: (0) nestlet intact, (1) flat nest with partially shredded material, (2) shallow nest with shredded material but lacks fully formed walls, (3) nest with well-developed walls, and (4) nest in shape of cocoon with partial or complete roof (Appendix A). 

Actigraphy was conducted as previously described [12]. Actigraphy is a sensitive, noninvasive technique that measures 24–7 activity levels. Briefly, rest-activity rhythms were assessed under standard lighting conditions in home-made Plexiglas chambers containing passive infrared sensors mounted on the underside of the lids. The dimensions of the transparent cylindrical Plexiglas chambers were 6-inch diameter × 10-inch height. Mice were individually housed during actigraphy with access to feed and water. Each gross movement of the animal was recorded as an activity count with VitalView acquisition software version 4.1 (Mini-Mitter Company, Inc., Bend, OR, USA). Activity counts were binned in 60-s epochs and scored on an activity scale (0–50) over a 3-day period. Data were analyzed with ACTIVIEW Biological Rhythm Analysis software version 1.3 (Mini-Mitter Company, Inc.). A chi-square periodogram method was used to determine the diurnal rest-activity period.

Feed intake was measured over a 4-day period. 2–3 mice of the same sex were housed per cage. The weight of the feed remnants in the cage on day 4 was subtracted from the weight of the feed added to the cage on day 1 and divided by the number of days and the number of mice per cage to calculate the average feed intake per mouse per day. 

Passive avoidance was conducted as previously described [12]. Passive avoidance is a fear-aggravated test used to assess learning and memory. Briefly, mice were acclimated to the experimental room for at least 20 min prior to testing in a foot shock passive avoidance paradigm using an aversive stimulator/scrambler (Med Associates Inc., Fairfax, VT, USA). A bench-top lamp was turned on behind the center of a light/dark shuttle box and aimed toward the back-left corner away from the dark side of the shuttle box. On the training day, a mouse was placed in the light side of the shuttle box toward the back corner away from the opening to the dark side of the shuttle box. The trap door in the shuttle box was open. After the mouse crossed over to the dark side, the trap door was closed and the latency time for the mouse to move from the light to the dark side was recorded. The mouse was allowed to equilibrate in the dark side for 5 s before receiving a 2-s 0.5 mA foot shock. After 15 s, the mouse was removed from the shuttle box and returned to its home cage. The apparatus was cleaned with 70% EtOH between animals. At test times (6 and 24 h after training), the mouse was placed in the light side of the shuttle box facing the left rear corner away from the opening to the dark side with the trap door open. After the mouse crossed to the dark side, the trap door was closed and the latency time for the mouse to move from the light to the dark side was recorded. If the mouse did not move to the dark side within 300 s, it was gently guided to the dark side and the trap door was closed. The mouse was allowed to equilibrate to the dark side for 5 s before returning to the home cage. Mice only received one shock on the training day. Testing at 24 h measured extinction.

Blood Collection and Analysis: Mice were anesthetized with isoflurane and blood collected from the inferior vena cava with a 21-gauge butterfly needle, mixed with sodium heparin anticoagulant, assayed for ketone and glucose levels, spun at 5000 rpm for 10 min, and the plasma layer quick frozen and stored at −80 °C. All samples were collected during the light phase between 8–10 am (nonfasted). Ketone and glucose levels were assessed in blood using a Precision Xtra blood glucose and ketone monitoring system (Abbott Diabetes Care Inc., Alameda, CA, USA). Plasma samples were submitted to the Biochemical Genetics Laboratory at the Wisconsin State Laboratory of Hygiene for quantitative plasma amino acid analysis.

Amino acids were quantified in blood plasma by ion-exchange chromatography on a Hitachi High-Technologies L-8900 Amino Acid Analyzer with column (PF High SPE, 6.0 × 40 mm, Hitachi 855-4515, Tokyo, Japan) as previously described [16] with minor modifications. Briefly, frozen plasma samples were thawed and 150 μL was mixed with 15 μL of 35% sulfosalicyclic acid (SSA) solution to achieve a 1:10 SSA to specimen ratio. Mixtures were vortexed for at least 10 s and spun at 14,000× *g* in a microcentrifuge for 3 min. The supernatants were filtered using a 0.22 μm PVDF syringe filter (Millex SLGVX13NL) and 1 mL disposable syringe. Equal volumes (typically 30 μL each) of eluant and internal standard solution (4 nmol aminoethylcysteine diluted in water) were vortexed, and at least 50 μL of this mixture was pipetted into a Hitachi sample vial (Hitachi ANO-2312) labeled with the specimen identification number and containing a 300 μL glass insert with poly-spring (Hitachi ANO-2313). Care was taken to avoid bubbles. In the case of bubbles, the insert was removed and briefly spun in a centrifuge to remove bubbles. The Hitachi vials were capped and loaded onto the IEC of the Hitachi Amino Acid Analyzer. Amino acids were selectively eluted from the IEC with buffers of increasing pH with a programmed method of varied flow rates and temperatures. After elution, ninhydrin was mixed with the buffer-amino acid solution, heated to develop the purple color and read at 570 nm for the amino acids phenylalanine and tyrosine. The total run time was 2.5 h per sample. The concentration of each amino acid was calculated by comparing the peak areas of the amino acid to the peak area of the internal standard, aminoethylcysteine, using Agilent OpenLAB software.

Degradation or major alterations can occur with long-term storage of amino acid specimens. For example, cystine will oxidize to cysteine disulfide(s) and glutamine will hydrolyze to glutamate. All samples were analyzed within 2 months from the date of collection.

Statistics: Mean data were plotted, and statistical significance determined by 2-way ANOVA with post hoc Tukey’s multiple comparison tests.

## 3. Results

### 3.1. C57BL/6J Growth in Response to Standard Diets

There were no differences in neuroassessments as an effect of diet (data not shown); all mice were healthy. There was a statistically significant 11–27% increase in body weight selectively in male mice fed Teklad 2019 or Purina 5015 compared to AIN-76A (Figure 2). Measurement of feed intake indicated increased intake of Purina 5015 compared to both Teklad 2019 and AIN-76A in both males and females (Figure 3) although there were no statistical differences in body weight in C57BL/6J mice fed Purina 5015 versus Teklad 2019.

These data indicate that standard diets can have large and sex-specific effects on growth metrics in wild type C57BL/6J mice. Correlation of body weight with actual feed intake remains to be determined. A limitation of the crude weighing method employed here to determine feed intake is that Purina 5015 is a much softer pellet diet than Teklad 2019 or AIN-76A, which could contribute to increased crumbling of the feed and the appearance of increased intake. Another limitation is that feed intake was measured at cage and not individual animal level. Regardless of these limitations, both chow diets significantly increased body weight selectively in male mice.

### 3.2. C57BL/6J Behavior in Response to Standard Diets

The effect of diet was tested in C57BL/6J mice with a behavior battery. There were neither significant differences in average total activity levels over the 24 h circadian cycle nor differences in nest building scores or passive avoidance-mediated learning and memory as an effect of the diet (Figure 4 and Appendix A). There was a significant decrease in nest height in male mice fed AIN-76A compared to Teklad 2019 (Appendix A), which did not significantly alter the overall nest score. Binning the actigraphy into light and dark quartiles indicated a potential 25% decrease in activity selectively in male mice fed Purina 5015 during the second half of the dark cycle, albeit the animal numbers were low because there was an equipment malfunction and animals associated with non-reliable data were removed from the analysis. Overall, the chow versus purified ingredient diet did not exert significant effects in wild type C57BL/6J mice in three common behavioral tasks.

### 3.3. Blood Plasma Biomarkers in C57BL/6J Mice in Response to Standard Diets

After completion of the behavioral battery test, blood was collected from euthanized mice and assessed for ketone, glucose and amino acid levels. There were no significant differences in plasma ketone or glucose levels in wild type C57BL/6J mice as an effect of diet (Figure 5).

There were significant differences in the levels of several amino acids in blood plasma as an effect of sex and diet (Figure 6 and Figure 7). Two-way ANOVA statistics are provided (Appendix A). Sex-specific differences were observed with decreased arginine and increased glutamate in males in response to Purina 5015 as well as decreased 1-methylhistidine and tryptophan in males in response to both AIN-76A and Purina 5015 (Appendix A). Diet-specific differences in females included increased isoleucine, leucine, methionine, proline, tyrosine and valine with AIN-76A (Appendix A), and in males increased glutamate, glutamine and ornithine with Purina 5015 (Appendix A). Diet specific differences in both sexes included decreased ethanolamine, glycine, 1-methylhistidine and taurine, and increased lysine and threonine with AIN-76A compared to one or both chows (Appendix A). Among the nine essential amino acids, isoleucine, leucine, methionine and valine were selectively elevated in females in response to AIN-76A (Appendix A). Lysine and threonine were elevated in both males and females in response to AIN-76A. The branched chain amino acids (BCAAs) isoleucine, leucine and valine were significantly higher in female mice fed AIN-76A compared to one or both chows (Appendix A).

## 4. Discussion

We observe increased body weight in male but not female wild type C57BL/6J mice maintained on rodent chow (Purina 5015 and Teklad 2019) compared to a purified ingredient diet (AIN-76A). The energy density of the diets is not the causal factor of increased weight gain in the male mice as Purina 5015 and AIN-76A have similar energy densities between 3.7–3.8 kcal/g, which is 12–15% higher than Teklad 2019 at 3.3 kcal/g. Regarding macronutrients, the quantity of protein in all three test diets was similar at 18–19%, albeit the source of the protein was different. The fat content of AIN-76A at 5.1% was about half of the chows at 9–11%, but higher fat alone is not sufficient to induce weight gain as a ketogenic diet with 8.6% protein, 75% fat, 3.2% carbohydrate and an energy density of 7.24 kcal/g significantly reduces weight gain in C57BL/6J male mice [12]. Likewise, powdered human Enfamil casein- and soy protein isolate-based infant formula diets formulated into pellets and fed to mice contain significantly less protein and higher fat than standard rodent diets, but are associated with reduced and equivalent growth curves, respectively, compared to Teklad 2019 [6]. Carbohydrate content alone is likely not sufficient to cause the reduced body weight in male C57BL/6J mice because AIN-76A has the highest and ketogenic diet has the lowest carbohydrate content and both are associated with reduced body weight. Whether the ratio of macronutrients or specific macronutrients cause the increased weight gain in male mice fed the chows remains to be determined.

A major difference between rodent chows and purified ingredient diets is the protein source. Purina 5015 primarily derives protein from ground wheat, dehulled soybean meal and ground corn; likewise, the major ingredients of Teklad 2019 are ground wheat, ground corn, corn gluten meal and wheat middlings. In contrast, purified ingredient diets such as AIN-76A contain casein as the protein source. Since both test chows elicit increased weight gain in wild type C57BL/6J male mice, the causal factor here is not soybean meal, which is specific to Purina 5015. A prior study comparing casein protein- and soy protein isolate-based diets formulated from powdered human infant formulas to Teklad 2019 indicates that growth curves for the soy cohorts mirror Teklad 2019 whereas the casein-based diet resulted in drastically reduced body weight despite significantly reduced protein and increased fat content in both infant formula diets [6]. Overall, these published and new data suggest that the type of protein or feed additives or contaminants in the diet affect weight gain. 

There is a controversy regarding the relative importance of protein quantity versus protein type on metabolic outcomes. Varied protein levels in rodent diets affect metabolic and molecular phenotypes in sex and genotype-specific manners in adult mice. Specifically, commencing a low protein diet (7% calories from whey protein) at 10 weeks of age in C57BL/6J male mice results in significantly reduced body weight by 7 weeks post-treatment compared to moderate (14%) and high (21%) isocaloric protein diets [17]. Protein source determines the potential of high protein diets to attenuate the development of obesity in C57BL/6J mice with casein as the most efficient in preventing weight gain and accumulation of adipose mass [18]. In our study, the calories provided by protein were similar at 18% for the Purina 5015, 23% for the Teklad 2019 and 19% for the AIN-76A, but body weight in males was selectively decreased in response to the casein-based AIN-76A. Others show a trend (*p* = 0.06) for increased weight gain in adult C57BL/6J male mice maintained for 3 weeks on casein-based purified ingredient diet compared to an isocaloric purified ingredient diet with wheat gluten, corn gluten and soy protein isolate as the protein sources (3.7 kcal/g energy value with 20% calories from protein) [19]. The trend for increased weight gain is lost at 8 weeks treatment. Major differences between that study and ours are: (1) we compared commonly used vivarium chows with a purified ingredient diet, and they compared two specialty-formulated isocaloric purified ingredient diets; (2) the Purina 5015 and Teklad 2019 chows contain soybean oil, AIN-76A contains corn oil, and the isocaloric purified ingredient diets contained corn and olive oils; and (3) we commenced treatment at 3 weeks of age versus their 9 weeks of age. Thus, it appears that both protein type as well as quantity affect weight gain in mice as an effect of sex and age with possible interactions with other ingredients such as fats. For example, high casein intake can accentuate, and high whey intake reduce (30% protein), the negative metabolic effects of a high fat diet [20].

In addition to decreased body weight, we observed altered blood plasma amino acid levels in response to purified ingredient diet. Of note, there was not a single amino acid that was selectively altered in male mice in response to both chows and not AIN-76A, which would have prompted future studies to directly examine the effect(s) of that amino acid on growth metrics. Glutamate, glutamine and ornithine were selectively elevated in male mice in response to Purina 5015, which could contribute to sex-specific differences but likely not the growth differences observed here as these amino acid levels were similar between Teklad 2019 and AIN-76A. The amino acids lysine and threonine were elevated in response to AIN-76A compared to both chows in both sexes, which inversely correlates to the decreased body weight in males but does not explain the lack of effect in females. 

Both chows contain ground wheat and ground corn; AIN-76A contains casein as the protein source. These proteins vary in their amino content resulting in altered postprandial amino acid availability. The majority of plant-based proteins are of lower quality containing reduced levels of essential amino acids including a shortage of leucine, lysine and/or methionine that may contribute to reduced anabolic capacity [21]. Relative to our study, wheat and corn contain significantly lower levels of lysine compared to casein [21]. Casein is enriched in the BCAAs isoleucine and valine compared to the plant-based proteins corn, wheat and soy [21]. We find elevated levels of isoleucine, leucine and valine in female mice fed AIN-76 compared to Purina 5015 and of isoleucine and valine in female mice fed AIN-76A versus Teklad 2019. We did not observe significantly altered plasma BCAA levels in male mice in response to diet. Others have observed that BCAAs are associated with obesity [22,23]. Here, we do not find a positive correlation between weight gain in mice and plasma BCAA levels.

Sex-specific differences have been observed in the regulation of amino acid metabolism in humans in response to exercise where amino acids become an important alternative fuel for men who have a greater reliance on carbohydrate than fat to fuel long duration exercise, but women do not need to draw on amino acids as an alternative fuel during beta-adrenergic blockade because of enhanced lipolytic responsiveness [24]. A lysine-supplemented diet is associated with increased satiety and decreased feed intake in male rats [25]. It is postulated that the intake of cereal grains with low lysine favors increased energy intake, which is associated with the development of obesity whereas the intake of dairy products favors lower energy intake and decreased body weight [25,26,27].

Both obesity and autism are current pediatric pandemics. The use of plasma amino acid ratios has been proposed as a method to both diagnose and treat autism [28,29]. Dysregulated amino acid metabolism was associated with a combination of elevated glutamine, glycine and ornithine and reduced levels of BCAAs in 17% of subjects with ASD from the Children’s Autism Metabolome Project (CAMP) [28]. Here, we find statistically significant reductions in the ratios of glutamine, glycine and ornithine to BCAAs in response to AIN-76A compared to Purina 5015 in both female and male mice. In addition, the ratios of glutamine and glycine to BCAAs were significantly reduced in females fed AIN-76A compared to Teklad 2019 (Appendix A). The clusters of glycine/isoleucine and ornithine/leucine identify 70% of the ASD subjects [29]. In this study, these ratios are significantly reduced in response to AIN-76A compared to one or both chows (Appendix A). Glutamate was found elevated in ASD [30], and here is elevated selectively in males fed Purina 5015 compared to both AIN-76A and Teklad 2019. Isoleucine was reduced in ASD and here is elevated in females fed AIN-76A compared to both chows. These data suggest that an amino acid profile associated with a casein-based diet and anti-obesity phenotype in mice is also associated with an anti-ASD phenotype in humans.

The implications of these findings are important for animal research and subsequent clinical trials. Diet can interact with drugs, for example, soy affects the pharmacokinetics and pharmacodynamics of the antiepileptic drug valproic acid [31]. Thus, diet/drug interactions in research animals maintained on varied single-source diets may help explain the low success rate of translating promising preclinical research findings into the clinic. Diet can also affect mouse behavior. For example, Dr. Stephen Maxson found reduced susceptibility to audiogenic-induced seizures (AGS) in C57BL/6J mice fed Rockland Mouse Breeder Chow versus Purina Laboratory Chow [32]. His colony in the Behavior Genetics Laboratory at the University of Chicago was switched from Purina Laboratory Chow to the Rockland Mouse Breeder Chow in January of 1961 while he was selectively breeding for high and low lines of AGS-susceptible mice. The diet was switched to maintain standard dietary conditions between facilities because Hoag and Dickie found increased survival in mice maintained on Rockland Mouse Breeder Chow (Diet #5; [33]), and in August of 1959, all colonies at Jackson Laboratories were switched to this diet. Rockland Breeder Chow is an open formula diet containing ground whole wheat, dried skim milk, corn oil and Brewer’s dried yeast with no animal fat [33]. A comparison diet with similar ingredient composition (Diet #3, [33]) contained animal fat and was associated with reduced litters per female and reduced average total number of mice weaned per female. The current dietary recommendation from Jackson Laboratories for maintenance of C57BL/6J is LabDiet 5K52 (19% protein, 6% fat, 3.2 kcal/g metabolizable energy density), which contains ground wheat, ground corn, wheat middlings, ground oats, fish meal, dehulled soybean meal, soybean oil, corn gluten meal and dehydrated alfalfa meal as the main ingredients. Thus, LabDiet 5K52 contains grain-based phytoestrogens and fish meal but no animal fat. Jackson Laboratories has no record of the use of Rockland Breeder Chow (personal communication). In comparison to the test diets used herein, the energy density of LabDiet 5K52 most closely resembles Teklad 2019, the fat content AIN-76A, and the soy-containing ingredients Purina 5015. The use of varied diets between laboratories and their frequent under-reporting likely contributes to different experimental outcomes. Potential effects of diet on epigenetics and the gut microbiome over multiple generations of breeding remain to be determined.

The limitations of this study included lack of indirect calorimetry to quantitate exact feed intake at the individual mouse level and the use of an older AIN diet formulation. The strengths of the study included assessment of both male and female mice in a battery of tests including weight anthropometrics, behavior and blood biomarkers. Future experiments can assess growth, behavior and blood-based biomarkers in control and transgenic mouse models of autism maintained on matched purified ingredient diets containing varied protein sources and oils to identify the bioactive component(s) contributing to body weight and behavioral phenotypes. It will also be important to measure energy intake through indirect calorimetry and long-term effects on epigenetics and the gut microbiome in response to diet. These types of studies could lead to a personalized medicine approach to therapy for persons with ASD and other disorders.

## 5. Conclusions

In conclusion, we find increased body weight in wild type C57BL/6J male mice in response to two common rodent chows, Purina 5015 and Teklad 2019, versus a purified ingredient diet, AIN-76A. The faster growth in response to chow cannot be attributed to altered energy densities of the diets. It remains to be determined how the percent macronutrients or specific nutrients in these diets differentially affect weight gain and amino acid balance in mice. These findings highlight the importance of transparent reporting of experimental details and their consideration during study design, which are vital for scientific rigor. Animals utilized for biomedical research are maintained on single-source diets in vivariums, and the choice of diet and other husbandry conditions can have significant effects on study outcomes. Altered diet-drug interactions may help explain why so many compounds that appear efficacious in preclinical research eventually fail in clinical trials. Concurrent studies in our laboratory are addressing the effects of protein source on weight gain, behavior and biomarker phenotypes in mice.

## Figures and Tables

**Figure 1 animals-12-03448-f001:**
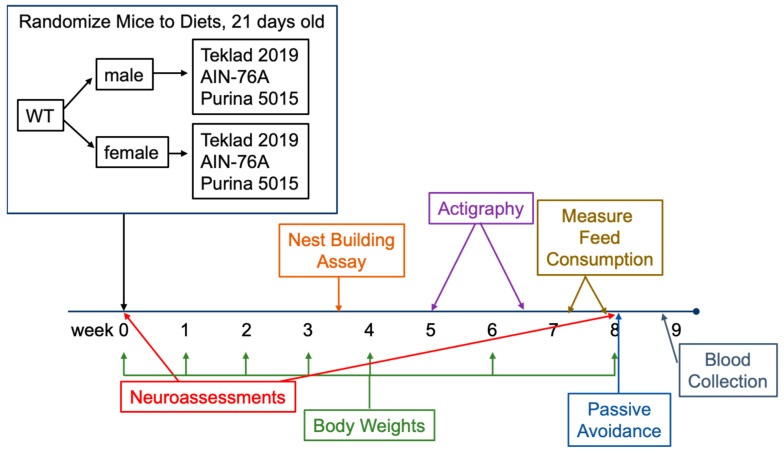
Study Design.

**Figure 2 animals-12-03448-f002:**
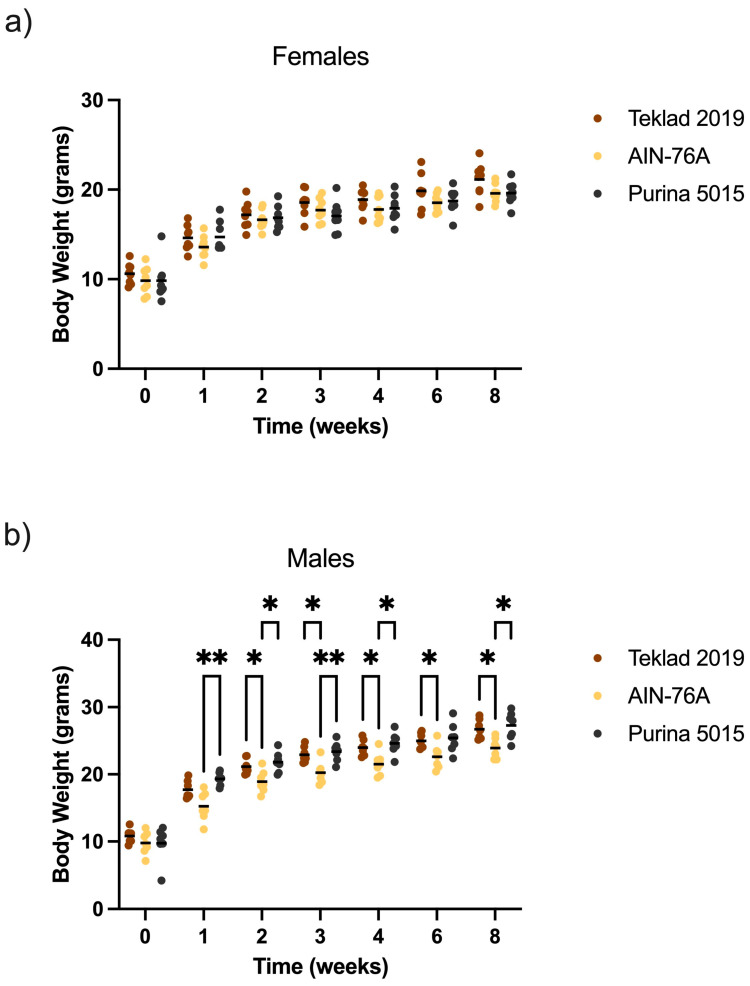
Mouse body weight as an effect of diet. Male and female C57BL/6J mice were weighed at age P21, randomized to diets (Teklad 2019, AIN-76A or Purina 5015), and weighed weekly for 4 weeks and biweekly thereafter. Body weight was plotted versus treatment time with diets. Statistics were determined by 2-way ANOVA and Tukey’s multiple comparison tests denoted by *p* < 0.05 (*) and *p* < 0.01 (**). (**a**) females (*n* = 8 per diet); ANOVA results: time × diet F(12, 126) = 2.115, *p* = 0.02; time F(2.474, 51.94) = 591.6, *p* < 0.0001; diet F(2, 21) = 1.402, *p* = 0.27; and subject F(21, 126) = 26.05, *p* < 0.0001. (**b**) males (*n* = 7 per diet). ANOVA results: time x diet F(12, 108) = 3.873, *p* < 0.0001; time F(2.111, 37.99) = 756.5 *p* < 0.0001; diet F(2, 18) = 7.924, *p* = 0.0034; and subject F(18, 108) = 16.87, *p* < 0.0001.

**Figure 3 animals-12-03448-f003:**
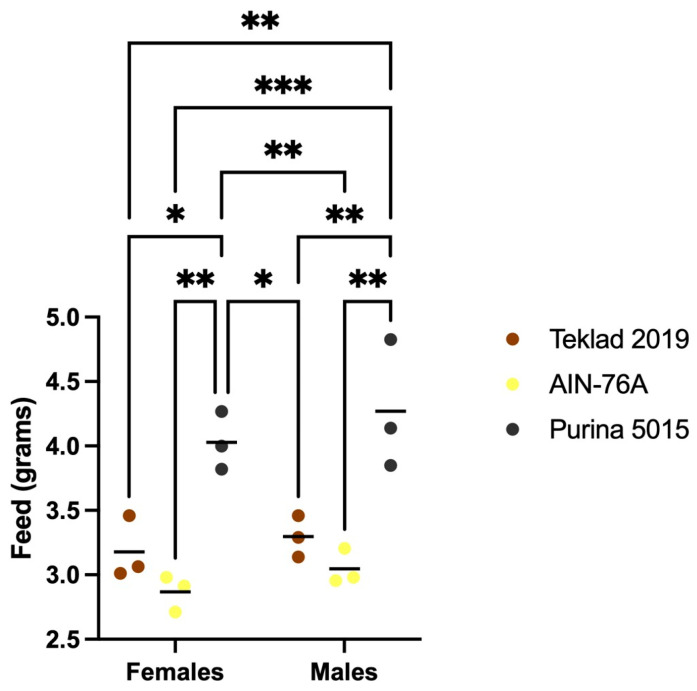
Mouse feed intake as an effect of diet. 2–3 mice of the same sex were housed per cage. Feed was weighed, added to the cage, and weighed again 4 days later. The average feed intake per mouse per day was plotted versus sex. Statistics were determined by 2-way ANOVA and Tukey’s multiple comparison tests denoted by *p* < 0.05 (*), *p* < 0.01 (**) and *p* < 0.001 (***). ANOVA results: interaction F(2, 12) = 0.0806, *p* = 0.92; sex F(1, 12) = 2.042, *p* = 0.18; and diet F(2, 12) = 32.75, *p* < 0.0001.

**Figure 4 animals-12-03448-f004:**
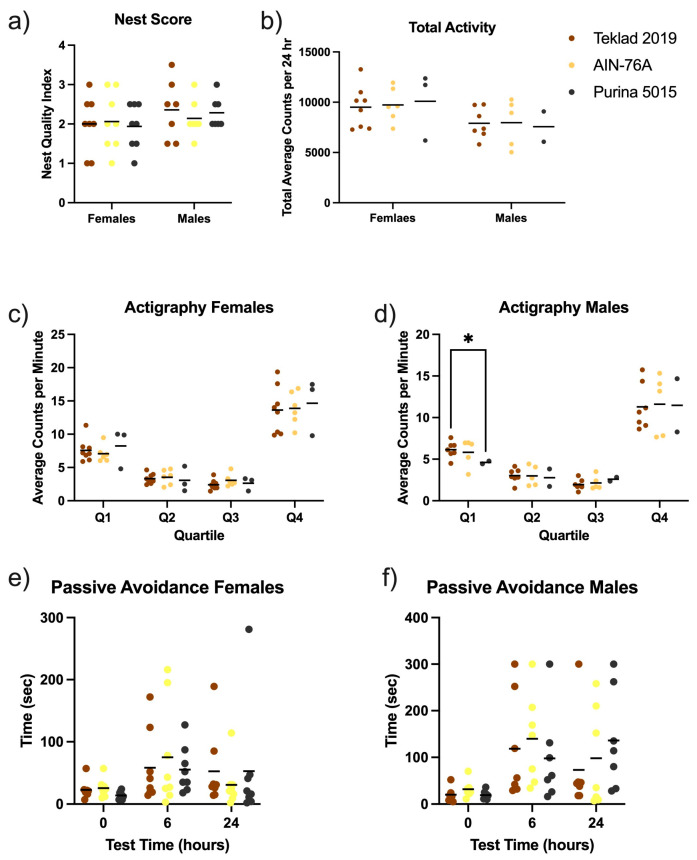
Mouse behavior as an effect of diet. Male and female C57BL/6J mice were tested in a behavioral battery including nest building, actigraphy and passive avoidance in response to diet (Teklad 2019, AIN-76A or Purina 5015). Statistics were determined by 2-way ANOVA and Tukey’s multiple comparison tests denoted by *p* < 0.05 (*). (**a**) nest score was plotted versus diet and sex. ANOVA results: interaction F(2, 39) = 0.2389, *p* = 0.79; sex F(1, 39) = 1.987, *p* = 0.17; and diet F(2, 39) = 0.066, *p* = 0.94. (**b**) total average activity counts per 24 h period plotted versus diet and sex. ANOVA results: interaction F(2, 25) = 0.089, *p* = 0.92; sex F(1, 25) = 5.436, *p* = 0.0281; and diet F(2, 25) = 0.017, *p* = 0.98. (**c**) average actigraphy counts per minute for female mice were binned into 6 h quartiles and plotted versus diet. ANOVA results: time × diet F(6, 42) = 0.4142, *p* = 0.87; time F(1.451, 20.31) = 177.6, *p* < 0.0001; diet F(2, 14) = 0.07965, *p* = 0.9239; and subject F(14, 42) = 4.362, *p* = 0.0001. (**d**) average actigraphy counts per minute for male mice were binned into 6 h quartiles and plotted versus diet. ANOVA results: time x diet F(6, 33) = 0.3313, *p* = 0.92; time F(1.270, 13.97) = 89.13, *p* < 0.0001; diet F(2, 11) = 0.03130, *p* = 0.9693; and subject F(11, 33) = 3.551, *p* = 0.0023. (**e**) passive avoidance times to enter the dark chamber for female mice were plotted versus diet. ANOVA results: time x diet F(4, 42) = 0.5027, *p* = 0.74; time F(1.707, 35.85) = 4.537, *p* = 0.0221; diet F(2, 21) = 0.02663, *p* = 0.97; and subject F(21, 42) = 1.619, *p* = 0.0909. (**f**) passive avoidance times to enter the dark chamber for male mice were plotted versus diet. ANOVA results: time x diet F(4, 36) = 0.8736, *p* = 0.49; time F(1.886, 33.94) = 11.31, *p* = 0.0002; diet F(2, 18) = 0.1725, *p* = 0.84; and subject F(18, 36) = 2.513, *p* = 0.0091.

**Figure 5 animals-12-03448-f005:**
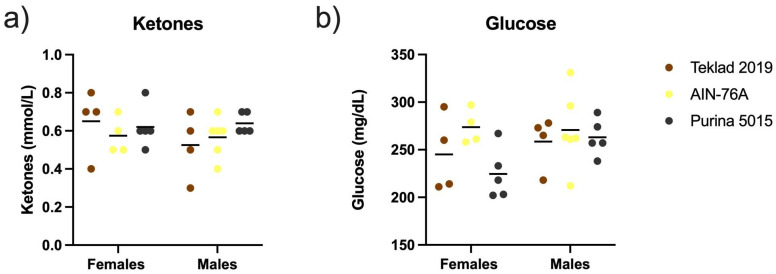
Plasma biochemistry as an effect of diet. Blood was collected from male and female C57BL/6J mice at euthanization and quantitated for plasma ketone and glucose levels as an effect of diet (Teklad 2019, AIN-76A or Purina 5015). Statistics were determined by 2-way ANOVA and Tukey’s multiple comparison tests with no significant differences. (**a**) ketones (*n* = 4–5 females and *n* = 4–6 males per diet); ANOVA results: interaction F(2, 22) = 0.8778, *p* = 0.43; sex F(1, 22) = 0.6742, *p* = 0.4204; and diet F(2, 22) = 0.6284, *p* = 0.54. (**b**) glucose (*n* = 4–5 females and *n* = 4–6 males per diet); ANOVA results: interaction F(2, 22) = 1.142, *p* = 0.34; sex F(1, 22) = 1.963, *p* = 0.18; and diet F(2, 22) = 2.251, *p* = 0.13.

**Figure 6 animals-12-03448-f006:**
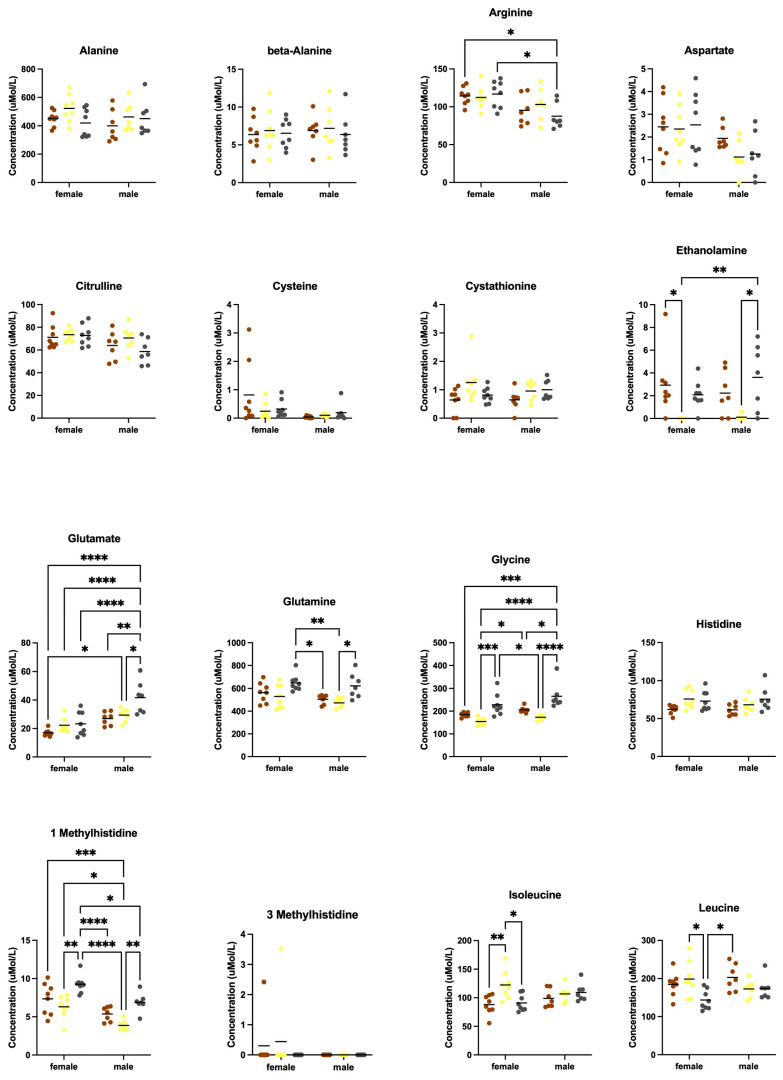
Blood plasma amino acid levels as an effect of diet. Blood was collected from male and female C57BL/6J mice at euthanization and quantitated for plasma amino acids by ion-exchange chromatography as an effect of diet (Teklad 2019, AIN-76A or Purina 5015). Statistics were determined by 2-way ANOVA and Tukey’s multiple comparison tests (see Appendix A) denoted by *p* < 0.05 (*), *p* < 0.01 (**), *p* < 0.001 (***) and *p* < 0.0001 (****). See Figure 7 for remaining data.

**Figure 7 animals-12-03448-f007:**
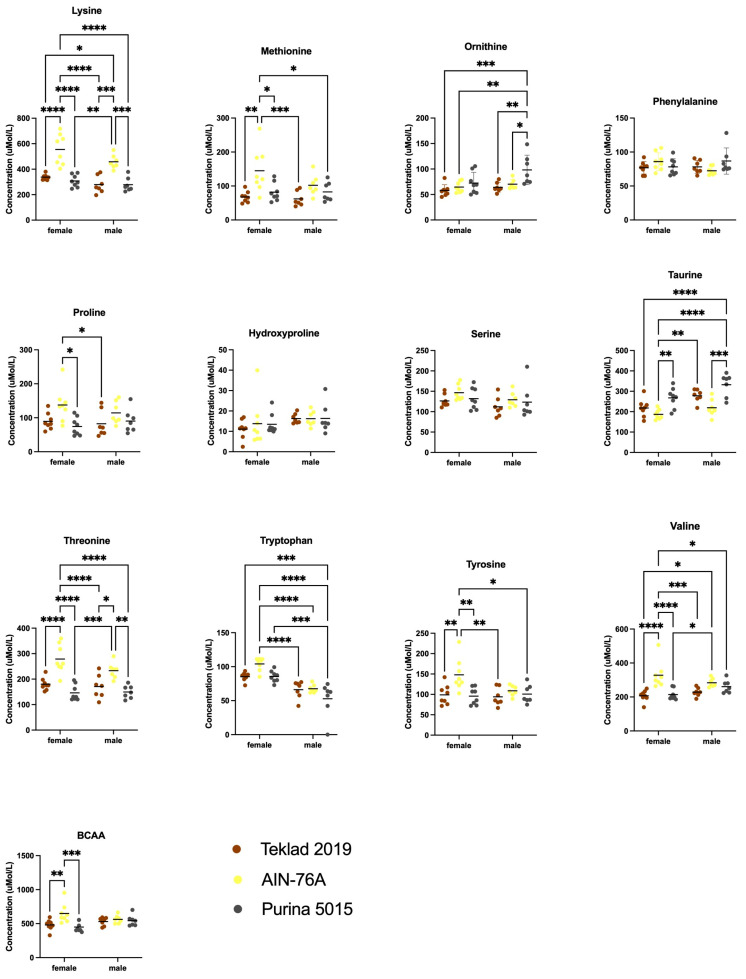
Blood plasma amino acid levels as an effect of diet. Blood was collected from male and female7C57BL/6J mice at euthanization and quantitated for plasma amino acids by ion-exchange chromatography as an effect of diet (Teklad 2019, AIN-76A or Purina 5015). Statistics were determined by 2-way ANOVA and Tukey’s multiple comparison tests (see Appendix A) denoted by *p* < 0.05 (*), *p* < 0.01 (**), *p* < 0.001 (***) and *p* < 0.0001 (****). See Figure 6 for remaining data.

**Table 1 animals-12-03448-t001:** Study Diets.

Diet	Energy Density (kcal/g)	Chemical Composition (%) (Protein, Fat, Carbohydrate)
Purina 5015	3.70	19, 11, 52
Teklad 2019	3.30	19, 9.0, 45
AIN-76A	3.79	18, 5.1, 65

## Data Availability

The data presented in this study are available within the article and Appendix A.

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
