# Peer review of "Improving Reproducibility to Enhance Scientific Rigor through Consideration of Mouse Diet"

_animals, 2022, doi:10.3390/ani12243448_

Round 1

Reviewer 1 Report

Westmark et al. aimed to compare the effect of three commonly used mouse diets on growth, behavior and blood biomarkers in male and female, wild type C57BL/6J mice.  They found significant differences in body weight and blood amino acid levels in the most commonly studied strain of mice. Overall, this study is well conducted and the outcomes are good. This article can be published at the present form.

Only a few minor concerns:

1. How did the authors determine the animal numbers?

2. The authors may discuss the strengths and limitations of this study.

Author Response

Thank you for the kind review. We have added comments to Track Changes in Word:

The number of mice tested was based on prior behavioral experiments.

The limitations of this study include lack of indirect calorimetry to quantitate exact food intake at the individual mouse level and the use of an older AIN diet formulation. The strengths of the study include assessment of both male and female mice in a battery of tests including weight anthropometrics, behavior and blood biomarkers. 

Reviewer 2 Report

Manuscript animals-2087764, entitled “Improving Reproducibility to Enhance Scientific Rigor through Consideration of Mouse Diet”

Recommendation:       The above paper is not suitable for publication in its present form.

The article provides useful information about the effects of mouse diet on the reproducibility of scientific research. Although, the experiment was in general appropriately designed and implemented and the article is well written there are some points that should be corrected or clarified.

L10: Which field?

L12: “For example” instead of “Of note”

L14 and throughout the text: “as an effect of” instead of “as a function of”

L15, 25: “found” instead of “find”

L20: “constitute an example highlighting” instead of “are an exemplar to illustrate”

L21: “…scientific rigor. In the present study, we examine the effects of three…”

L26: “We did not…”

L27: “…mice. However, amino acid profiles changed as an effect of sex…”

L33: “Transparent experimental variables…”

L45-46: Please rephrase

L49: “function” instead of “translate”

L51-52: Please delete

L56: “…mouse diet. Since multiple…”

L70: “The researchers of our laboratory have found…”

L98 and throughout the text: “feed” instead of “food”

L120-121: The part of the statistical analysis should be removed in a separate section at the end of Material and Methods and before the Results.

L139: “…or complete roof (Fig. S2).”

L151-152: “Feed intake was measured over a 4-day period. 2-3 mice of the same sex were housed per cage.”

L152: “feed remnants” instead of “food remaining”

L154-155 and throughout the text: “feed intake” instead of “food consumption”

L156: “…was conducted as previously described [12].”

L213-216: Repetition. Not necessary here. Please delete

L218: Please delete “versus female”

L229: “…at cage and not individual…”

L251: “Mouse feed intake as an effect of diet. 2-3 mice of the same sex were housed per cage”

L258: Please rephrase

L258-261: This part should be removed in material and methods. In detail, L258-259 in L131, L259-260 in L140 and L260-261 in L156

L261: “neither” instead of “no”

L263-264: “…and memory as an effect of the diet…”

L268-269: Please be precise. This malfunction could lead to non-reliable results

L296: “After completion of the behavioral battery test, blood…”

L310: “…provided (Table S1). Sex-specific differences were observed with decreased…”

L318: “…chows (Figure S8). Among the nine essential…”

L438: “were” instead of “are”

Author Response

Thank you for the thorough review of our paper. We addressed the comments in Track Changes in Word.
